# Somatic mutation distribution across tumour cohorts provides a signal for positive selection in cancer

Martin Boström [1] & Erik Larsson [1] ✉

Cancer gene discovery is reliant on distinguishing driver mutations from a multitude of passenger mutations in tumour genomes. While driver genes may be revealed based on excess mutation recurrence or clustering, there is a need for orthogonal principles. Here, we take advantage of the fact that non-cancer genes, containing only passenger mutations under neutral selection, exhibit a likelihood of mutagenesis in a given tumour determined by the tumour's mutational signature and burden. This relationship can be disrupted by positive selection, leading to a difference in the distribution of mutated cases across a cohort for driver and passenger genes. We apply this principle to detect cancer drivers independently of recurrence in large pan-cancer cohorts, and show that our method (SEISMIC) performs comparably to traditional approaches and can provide resistance to known confounding mutational phenomena. Being based on a different principle, the approach provides a much-needed complement to existing methods for detecting signals of selection.

Distinguishing somatic driver mutations from a multitude of passenger mutations is a central problem in cancer genomics[1,2]. Due to positive selection, driver mutations will tend to occur more frequently than expected by chance, and the main principle used by tools for detecting driver mutational events is thus to find genes with an excess of protein-altering or otherwise impactful mutations, relative to a background model or in relation to non-functional or low-impact mutations[3–6]. Some tools will also consider positional biases in gene mutation patterns, which may be indicative of gain-of-function alterations affecting a specific residue or protein domain[7,8]. The same principles can be applied to non-coding genomic elements[9].

While initially plagued by false positives, these frequency/recurrence-based approaches have been gradually refined and successfully applied to uncover a large number of driver genes in a wide range of cancers[10]. Consideration of mutation rate covariates such as replication timing and gene expression has been key to improving the results[4]. Still, remaining imperfections in the models, in combination with large cohorts and high statistical power, means that false positives remain a problem. Localised mutational phenomena, such as site-specific AID or APOBEC mutagenesis or UV-induced DNA damage hotspots at transcription factor binding sites, can have a particularly confounding effect on frequency-based methods[11–13]. Furthermore, many driver mutations are believed to reside in the long tail of relatively infrequent events where both sensitivity and specificity can be an issue, leaving many potential signals in a grey zone of unclear relevance[14]. Altogether, there is a pressing need for alternative principles that can complement existing methods by providing orthogonal evidence of selection.

In this work, we propose an alternative to frequency statistics, based not on the sum but instead on the distribution of mutated cases across a cohort for a given gene or genomic element; information that is typically disregarded. In essence, for genes containing only passenger mutations, the likelihood of mutagenesis in a given tumour is determined simply by the activities of the mutational processes active in that tumour. Gene mutation status will therefore correlate positively with mutation burden across samples, all other things being equal. However, in the presence of selection, we expect this correlation to be disrupted, since the selective force acts independently of the

[1]Department of Medical Biochemistry and Cell Biology, Institute of Biomedicine, Sahlgrenska Academy, University of Gothenburg, SE-405 30 Gothenburg, Sweden. ✉e-mail: erik.larsson@gu.se

mutational processes. As an example, we have previously shown how mutations at genomic hotspots hypersensitive to UV mutagenesis are, as expected, strongly correlated with tumour mutation burden in melanoma, whereas known drivers deviate from this pattern[15]. We here formalise this principle for systematic discovery of cancer driver mutations, and show that this approach, which is applicable to both coding and non-coding regions, performs comparably to existing methods and can be less sensitive to flaws in the background model. As the method is not based on recurrence, it provides a useful complement that may help strengthen or weaken the verdict of existing tools.

## Results

### Identifying signals of positive selection from cohort mutational skew

We implemented a method (SEISMIC) that, for a given gene or region, evaluates whether the observed distribution of mutated vs. non-mutated tumours across a cohort deviates from the expected pattern under neutral selection (Fig. 1; detailed description in "Methods"). Briefly, for each gene and tumour, the probability of mutagenesis is estimated while taking into account tumour mutational burden, gene sequence and trinucleotide signature[16]. The latter is determined per tumour in the case of whole genome sequencing (WGS) data, or per cancer type in the case of whole exome sequencing (WXS) due to lower mutation counts. Likelihood scores are used to statistically assess whether an observed outcome (cohort mutation pattern) deviates relative to a large number of simulated outcomes, which are drawn from the estimated probabilities while calibrating the recurrence to be the same as the observed data (Fig. 1). Information normally lost when mutations are summarised into counts is thus taken advantage of while mutation frequency is disregarded. Intuitively, for a given gene, the method tests whether mutations are unexpectedly skewed towards samples with lower mutation probability; typically those with lower mutation burdens.

### Cancer genes exhibit cohort mutational skew in melanoma

As an initial test, we applied our method to coding sequences (CDS) in cutaneous melanoma, based on 466 WXS samples from The Cancer Genome Atlas (TCGA). These samples spanned a wide range of mutational burdens, from 1 to 18,701 single nucleotide variants (SNVs) per

exome and were expectedly dominated by a trinucleotide profile consistent with mutagenesis by UV light (Supplementary Fig. 1).

We found six statistically significant genes at a false discovery rate of 5% ($q < 0.05$; BRAF, NRAS, MAP2K1, PTEN, GNAQ and KIT), all of which are known cancer genes[10]. For these genes, the cumulative distribution of mutated samples across the cohort deviated strongly from the expected selectively neutral distribution based on simulations, with $q$-values reaching as low as $3.3 \times 10^{-90}$ for BRAF (Fig. 2a, Supplementary Data 1). The samples carrying non-synonymous mutations in these genes were relatively uniformly distributed across the cohort with no obvious correlation with sample-specific mutational burden, which essentially is proportional to mutation probability in our model (Fig. 2b). This stood in sharp contrast to the likely passenger genes TTN, PCLO, DNAH7 and MUC17[6] for which the cohort mutation distribution was in close agreement with the simulated neutral outcomes (Fig. 2a; $q > 0.97$), with strong aggregation of mutated cases in the high-burden range and near-absence among low-burden tumours (Fig. 2b).

WGS data from 183 independent cutaneous melanomas[17] confirmed strong correlation between sample mutation burden and non-synonymous CDS mutations for the non-cancer genes as well as deviation from this pattern for the same set of driver genes (Fig. 2c). When repeating the analysis for the same genes but based on intronic mutations, which are generally not under positive selection, all of the genes instead exhibited passenger-type mutation distributions across the cohort as expected (Fig. 2c). The same passenger pattern was seen for a larger set of randomly selected non-cancer gene census (CGC) genes (Supplementary Fig. 2). SEISMIC thus revealed relevant cancer genes in melanoma, which exhibited a distribution of mutated cases across the cohort that deviated strongly from the neutrally expected patterns, while passenger genes showed patterns highly consistent with simulated neutral selection.

Four of the six genes (BRAF, NRAS, PTEN, MAP2K1) are well-known cutaneous melanoma drivers[18] and were also revealed by frequency-based methods when applied to the same WXS data (dNdScv[5] and MutSigCV;[4] Supplementary Data 2, 3). The remaining two, GNAQ and KIT, had the lowest mutation frequencies of the six genes (Fig. 2b) and were not significant with these methods, but are both known drivers linked primarily to non-UV-exposed melanoma subtypes. GNAQ

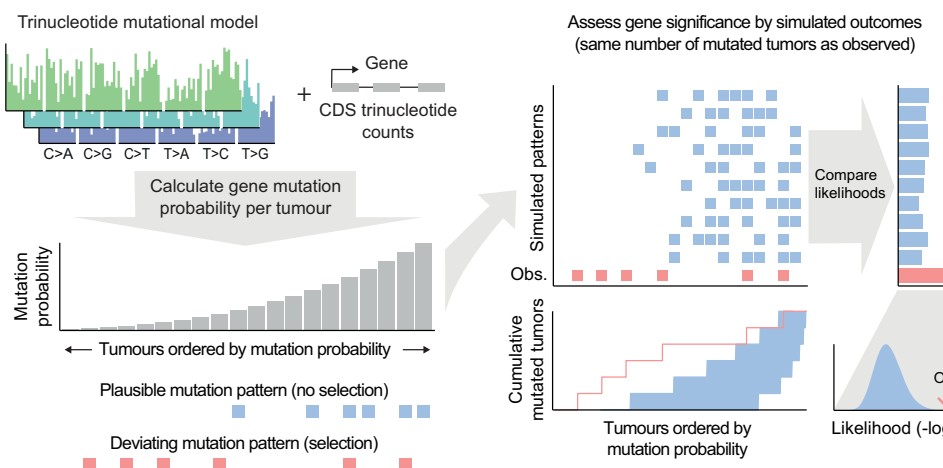

**Fig. 1 | Detection of positive selection based on skewed mutation distribution across cohorts.** In the SEISMIC method, the probability of mutagenesis for a given gene and tumour in a cohort is estimated based on trinucleotide signatures (per-sample or per-cohort) and tumour-specific burdens (left side). Genes under neutral selection are assumed to exhibit patterns of mutagenesis in agreement with these probabilities across the cohort (blue squares), whereas genes under selective pressure for mutations are expected to show deviating patterns (red squares).

Significance is assessed by comparing the observed outcome for a given gene (pattern of mutated tumours across the cohort) to multiple simulated outcomes based on the estimated neutral probabilities, and differences are visualised by plotting the cumulative number of mutated tumours across the cohort ordered by gene mutation probability (CMT plot; right side). By default, only missense and nonsense mutations are considered, as synonymous mutations are assumed to generally be selectively neutral. CDS coding sequence.

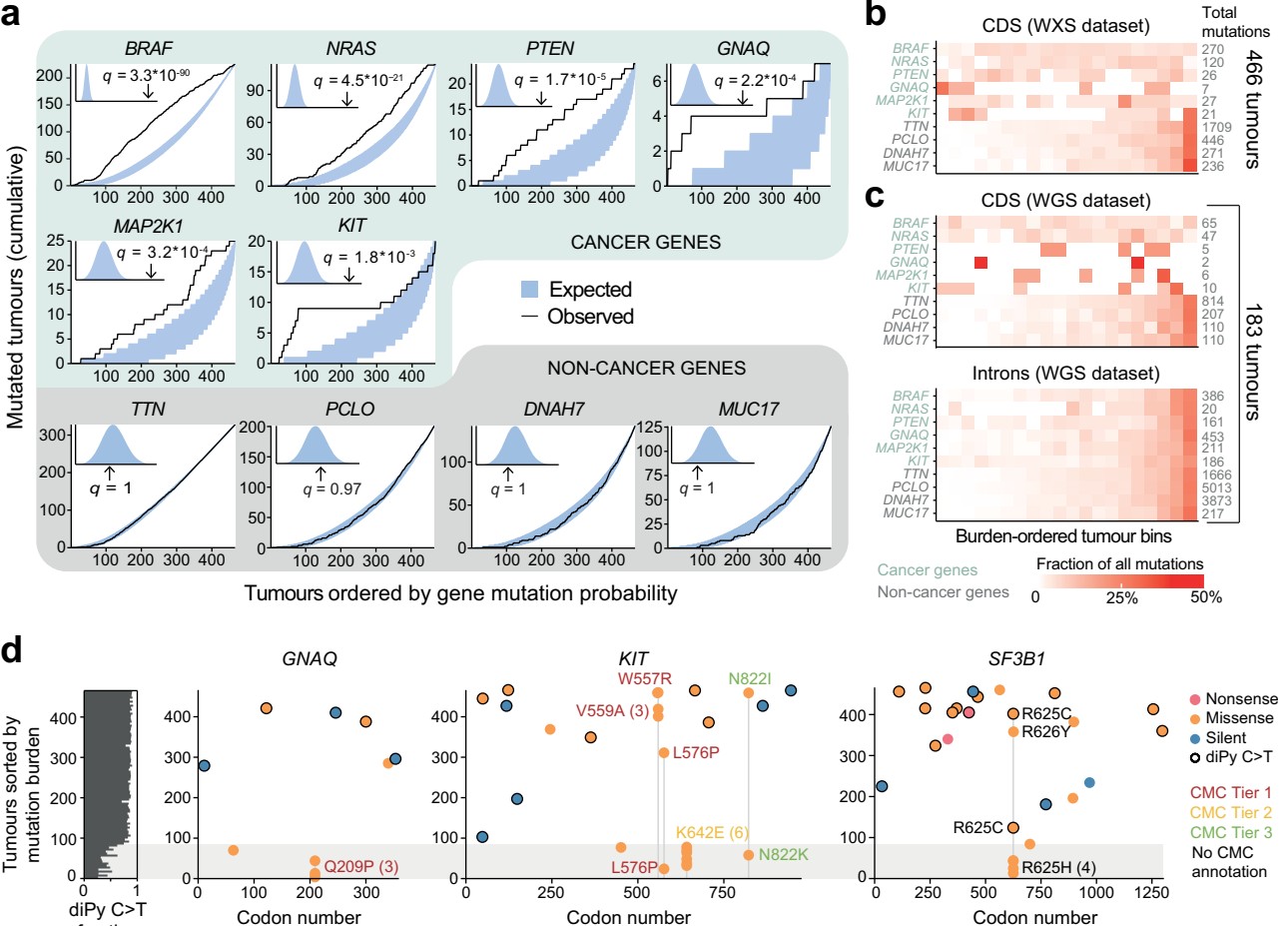

**Fig. 2 | Cancer genes identified by SEISMIC in a skin melanoma cohort.**
**a** Cumulative mutated tumours plot (CMT plot, see Fig. 1) for six significant cancer genes identified by SEISMIC at a false discovery rate ($q$) <0.05 (see "Methods"), plus non-cancer gene examples, in melanoma WXS data (CDS regions only; $n = 466$ tumours). The blue area represents the least extreme 90% of simulation outcomes. Inset: Gamma distribution fit of the likelihoods of the simulated cohorts in blue vs. that of the actual cohort, with $q$-values indicated. **b** Heatmap of non-synonymous mutations in tumours ordered by mutation burden (here equivalent to gene mutation probability) and binned, for the genes from (**a**). In non-cancer genes, the number of mutations in each bin increases with mutation burden, but this correlation is disrupted in the cancer genes. **c** Same as in (**b**) but repeated using melanoma WGS data, with CDS and intron-based results shown separately. **d** Positionally clustered known driver mutations in *GNAQ*, *KIT* and *SF3B1* occurring preferentially in non-UV-exposed samples. Tumours are sorted by burden along the y-axis, with the proportion UV-type mutations (C > T in dipyrimidine contexts) indicated. Known driver mutations in Cancer Mutations Census (CMC) are indicated, with Tier 1 having the strongest evidence. CDS coding sequence, diPy dipyrimidine, WGS whole-genome sequencing, WXS whole-exome sequencing. Source data are provided as a Source Data file.

mutations, which are common in uveal melanoma, have also been found in mucosal melanoma[17,19,20]. *KIT* mutations, while also found in skin cutaneous melanoma, are more prevalent in mucosal and acral melanoma[17].

Relative to the other genes, we found that *GNAQ* and *KIT* exhibited elevated mutations in low-burden samples (Fig. 2b), which were lacking the traditional UV mutational signature (C > T transitions in dipyrimidine contexts arising from cyclobutane pyrimidine dimer formation[16,21]; Fig. 2d). Several of these mutations exhibited strong positional clustering while also being annotated in the Cancer Mutation Census (CMC)[22] as known driver mutations, thus further supporting a driver role (Fig. 2d). Notably, positionally recurrent *GNAQ* mutations (Q209P, $n = 3$) occurred exclusively in the non-UV range of the cohort, consistent with *GNAQ* being a driver in non-UV melanomas[17,19,20]. *KIT* similarly exhibited positionally recurrent mutations preferentially in the non-UV samples (K642E, $n = 6$) but also more rarely in UV-exposed tumours (primarily V559A, $n = 3$), in agreement with infrequent *KIT* driver mutations in regular cutaneous melanomas[17], seemingly with different mutations being relevant in different subtypes.

Similar results were obtained for *SF3B1*, another cancer gene commonly mutated in uveal melanoma and previously identified as a driver gene in mucosal melanoma[17]. While not in the initial list of significant genes ($q = 0.13$), *SF3B1* nevertheless exhibited skew toward low-burden tumours (Supplementary Fig. 3) with all mutations in non-UV samples being identical (R265H, $n = 4$; Fig. 2d). Three additional clustered mutations were also found among UV-exposed samples (R625C and R626Y). Notably, all mutations in non-UV samples for *SF3B1*, as well as *GNAQ* and *KIT*, were non-synonymous, in further support of positive selection.

Taking these results together, the approach pinpointed well-established driver genes also detected by frequency-based methods, but also likely drivers not readily detected by such methods, giving further support for *GNAQ*, *KIT* and *SF3B1* driver mutations in cutaneous melanoma. A key feature of these mutations was preferable occurrence in tumours more reminiscent of other subtypes than the UV-exposed melanomas that dominated the cohort.

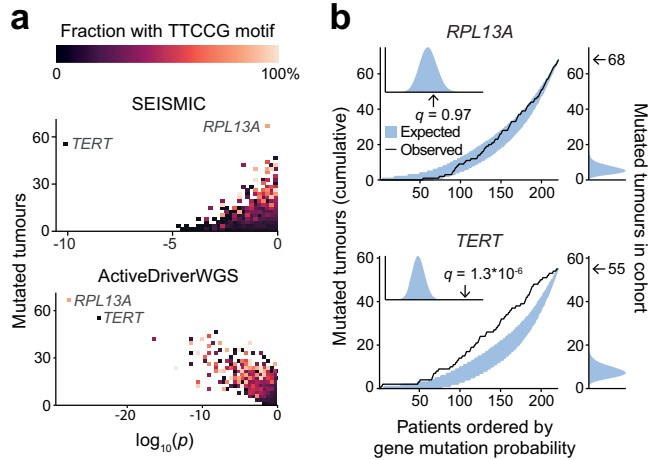

**Fig. 3 | Resilience to confounding effects from ETS-related local hotspots in melanoma promoters. a** Significance (uncorrected *p*-value) vs. mutational recurrence for promoters (500 bp upstream regions) in melanoma (*n* = 221 tumours), for the SEISMIC method (based on cohort mutational skew; top) and ActiveDriverWGS (frequency-based; bottom). The proportion of mutations within 10 bp of a TTCCG sequence is indicated for each promoter, in order to pinpoint confounding recurrent mutations due to increased UV damage susceptibility at ETS transcription factors binding sites. Promoters were binned to avoid overplotting. Both tests correctly identify *TERT* promoter mutations as drivers, but SEISMIC also lacked enrichment of TTCCG-related hotspot mutations among top hits. **b** CMT plots (see Fig. 1) for the *RPL13A* and *TERT* promoters, with observed and expected recurrence on the right side. Unlike *TERT*, mutated cases with respect to the *RPL13A* promoter fall closely within the simulated expected distribution across the cohort, thus avoiding significance despite strong excess burden. Source data are provided as a Source Data file.

## Resilience to confounding localised mutagenesis in non-coding regions

Next, we applied the same principle to non-coding promoter regions in melanoma. These are challenging due to presence of numerous localised UV damage hotspots arising at transcription factor binding sites; primarily TTCCG sequences relating to E26 transformation specific (ETS) family factors[15,23,24]. These are associated with local mutation hotspots that have a strong confounding effect on existing methods for detecting signals of positive selection, since expected mutation rates are vastly underestimated at these sites[11,13,25].

When analysing 500 bp upstream regions using melanoma WGS data, we found that the *TERT* promoter stood out as highly significant ($q = 1.3 \times 10^{-6}$), and that TTCCG-related hotspot mutations were markedly absent among the top results using our approach (Fig. 3a, Supplementary Data 4). Promoter mutations in *TERT*, arguably the most well-established non-coding somatic driver mutations described to date[11,26,27], were likewise readily identified based on frequency by ActiveDriverWGS[9] (Fig. 3a, Supplementary Data 5; $q = 9.4 \times 10^{-21}$). However, as seen with other existing methods[13], large numbers of recurrent TTCCG-related mutations were also among the top-scoring results, notably with the *RPL13A* promoter surpassing even *TERT* in terms of recurrence and significance. *RPL13A* is known to harbour a particularly strong UV hotspot verified in cell culture to be highly susceptible to UV mutagenesis[15].

*RPL13A* and *TERT* serve to illustrate the difference between the approaches. While both promoters were recurrently mutated at a much higher degree than expected, SEISMIC ignores recurrence and assesses significance relative to simulated outcomes having the same frequency as the observed data. With this model, mutated cases with respect to the *RPL13A* promoter across the cohort closely followed the expected selectively neutral distribution, while *TERT* showed

significant skew toward low-burden tumours (Fig. 3b). Under the assumption that the two principles are vulnerable to different types of confounding effects, we combined the methods through selection of the most conservative *p*-value for each promoter, which led to an even better separation between *TERT* and the other genes (Supplementary Fig. 4). Our approach thus readily identified relevant non-coding promoter mutations while proving resistant to false positive signals that have plagued other methods in melanoma.

In an attempt to find non-coding drivers other than the *TERT* mutations, we tested promoters in 22 individual cancer types, as well as in a large combined pan-cancer cohort, using WGS mutation calls from the Pan Cancer Analysis of Whole Genomes (PCAWG) consortium (2,658 tumours)[25]. After correcting for multiple testing, the only significant promoters were *TERT* in cutaneous melanoma and pan-cancer; *BCL7A*, *IGLL5*, *BCL2*, and *POLR3E* pan-cancer; and *HNRNPR* in uterine adenocarcinoma (Supplementary Data 6). *BCL7A*, *IGLL5*, and *BCL2* are likely false positives as they were mutated primarily in lymphomas, where they are known targets of local AID-mediated somatic hypermutation[13]. Interestingly, *POLR3E* promoter mutations were also pinpointed based on frequency statistics in the same dataset[13], and our results thus further support that these may be under positive selection.

## The SEISMIC principle identifies cancer drivers across cancer types

Following our initial evaluation of the methodology, we applied SEISMIC to 14 individual cancer types as well as a large pan-cancer cohort using WXS data from TCGA (see "Methods"). This identified 156 significant signals ($q < 0.05$) in 12 cancers and pan-cancer, 69% of which were canonical cancer genes according to CGC[22], representing 99 unique genes exhibiting mutational skew in at least one cohort (Fig. 4a, Supplementary Data 1). In most of the significant genes, we note that individual mutations annotated as putative or known driver events in CMC exhibited greater mutational skew than other mutations, consistent with these being under positive selection (Supplementary Fig. 5, Supplementary Data 7).

Of the analysed cancer types, uterine corpus endometrial carcinoma (UCEC) showed the largest number of significant genes, as well as a high proportion of hits listed in CGC or identified in other studies[4–6,28] (Fig. 4a). This is in agreement with UCEC having a relatively large number of mutational drivers[5], but statistical power may also contribute as this cancer type showed the highest mean number of mutated tumours per gene (Supplementary Fig. 6). Notably, inclusion of many hypermutated tumours, which may otherwise present a challenge for driver detection tools[29], contributed positively to SEISMIC's ability to discriminate drivers from passengers in this cohort (Supplementary Fig. 7, Supplementary Data 1, 8–10).

We contrasted our results in UCEC with those from MutPanning[6], dNdScv[5], and MutSigCV[4] applied to the same dataset (Supplementary Data 11–13), and found that our method showed comparable enrichment of known cancer genes (Fig. 4b). Comparing the genes identified by the different methods, we note that any combination of methods yielded a more conservative list of high-confidence cancer genes (Fig. 4c). The dNdScv, MutPanning, and SEISMIC methods all individually identified CGC genes that did not reach significance in any of the other methods (Fig. 4c).

In most cancer types, there were considerable overlaps between our results and genes identified in other studies[4–6,28], but also hits with limited or no support in the other catalogues (Fig. 4a). While these should be treated with caution, some were plausible true positives. These included the protein kinase *DCLK1* in stomach adenocarcinoma (STAD), which has previously been reported as a putative driver in gastric cancer[30]. *TLR4* was also identified in STAD; a gene proposed to contribute to gastric cancer progression by mediating an inflammatory response to bacterial lipopolysaccharides (LPS)[31]. Some of the *TLR4* mutations clustered at positions involved in LPS recognition[32]

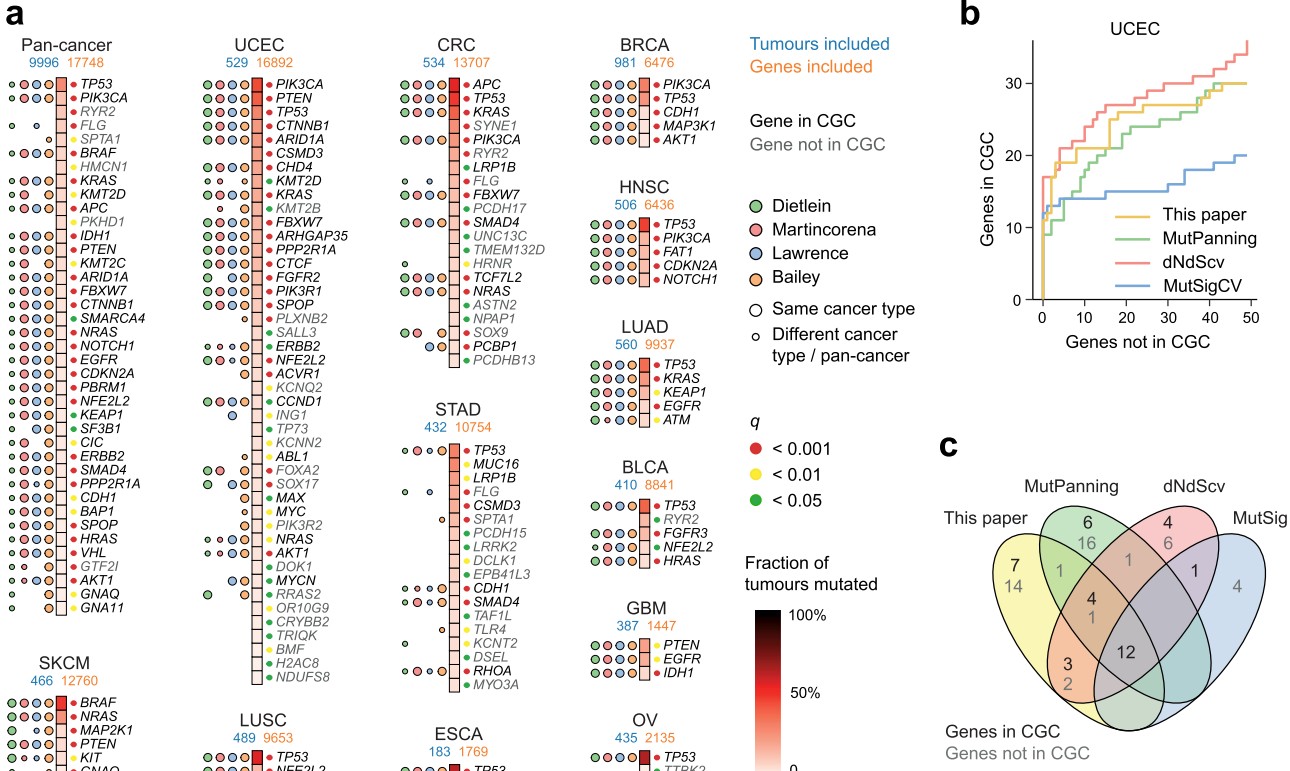

**Fig. 4 | Driver genes identified by low-burden skew. a** Genes exhibiting significant cohort mutational skew in 14 cancer types as well as pan-cancer (LIHC and CESC lacked significant results). Results are ordered by recurrence, with coloured dots on the right side indicating level of significance, and on the left side indicating support in other driver studies in the same/different cancer type[4–6,28]. Canonical cancer genes (from the Cancer Gene Census, CGC) marked with black text. Genes with ≥3 mutations were considered in each cancer. **b** Cumulative plot showing enrichment of canonical cancer genes among the most significant genes using our method, MutPanning, dNdScv, and MutSigCV, on the same UCEC dataset. **c** Results from (**b**)

shown as a Venn diagram of overlapping significant genes ($q < 0.05$, false discovery rate). The number of CGC genes is indicated in each set. BLCA bladder carcinoma, BRCA breast carcinoma, CESC cervical carcinoma, CRC colorectal carcinoma, ESCA oesophageal carcinoma, GBM glioblastoma, HNSC head and neck carcinoma, LIHC liver hepatocellular carcinoma, LUAD lung adenocarcinoma, LUSC lung squamous cell carcinoma, OV ovarian adenocarcinoma, SKCM cutaneous melanoma, STAD stomach adenocarcinoma, UCEC endometrial carcinoma. Source data are provided as a Source Data file.

(Supplementary Figs. 8, 9), which is in agreement with an earlier TCGA-based study that demonstrated a modulatory effect on LPS signalling for several such mutations[33]. *DOK1*, pinpointed by our method in UCEC and encoding a RasGAP-binding protein involved in apoptosis[34], has been reported as a candidate tumour suppressor in several cancer types[35–37]. *ING1*, a pro-apoptotic tumour suppressor[38], showed a tumour suppressor-like profile with 4 positionally clustered nonsense mutations in low-burden UCEC tumours, superimposed onto additional mutations exhibiting a more passenger-like cohort distribution and substitution type characteristics (Supplementary Fig. 10). The pro-apoptotic BCL-2 family member *BMF*[39] was mutated in 4 UCEC patients (0.76%) of which 3 were low-burden tumours with nonsense mutations, similarly suggestive of a tumour suppressor role (Supplementary Fig. 11). Low levels of recurrence may render such putative drivers difficult to detect with frequency-based approaches.

SEISMIC also reported several large cancer genes not detected in the other studies, including *CSMD3* (3,707 amino acids) in UCEC and STAD, and *LRP1B* (4599 amino acids) in STAD and colorectal cancer (CRC; Fig. 4a). A low driver-to-passenger mutation ratio in large genes could make selection signals challenging to detect using traditional methods, while SEISMIC would be more sensitive to a relatively small number of driver events if these deviate markedly from the modelled cohort distribution. While caution is warranted as these two CGC genes have previously been suggested to represent false positives[4], our results may justify further investigation of putative driver roles.

With WXS data, SEISMIC uses a simplified mutation model where trinucleotide signatures are determined per cancer type rather than per tumour. In analyses of single cancers, this essentially makes the estimated mutation probability for a given gene and tumour proportional to the tumour's mutation burden. Exome-based analysis of PCAWG WGS mutation calls allowed us to compare the two models using the same data. While significant hits expectedly were reduced compared to the larger WXS-based analysis, we found that results based on the simplified (WXS) model correlated strongly with those from the patient-specific (WGS) model in most cancer types (Supplementary Fig. 12, Supplementary Data 14, 15), with little difference in cancer gene enrichment among top-scoring genes (Supplementary Fig. 13). In analyses of coding regions, the increased statistical strength provided by currently available WXS cohorts may thus often outweigh the benefits of more granular mutational modelling in WGS data.

## Discussion

Here, we establish that signals of selection can be effectively revealed based on skewed mutation distribution across cancer cohorts. Importantly, this can be achieved while disregarding mutation frequency/recurrence, which is the fundamental metric normally used to assess selection; a deliberate feature of our method with the aim of providing independent support compared to existing approaches. While we consider the difference in fundamental principle more important than absolute performance, we also find that the method performs well compared to existing tools when applied to diverse

cancer cohorts. Driver detection based on cohort mutational skew can thus form an important complement to the existing toolbox, by nominating additional candidates as well as providing orthogonal support for existing candidates. The search for weaker drivers on the edge of statistical significance may be particularly well served by verdicts from independent approaches.

While frequency/recurrence is an informative metric that could easily be considered in our model for increased power, our analysis of melanoma promoters exemplifies how scoring independently of recurrence can be beneficial by providing resistance to mutational model inaccuracies that confound frequency-based methods. The approach can still be vulnerable to model flaws; in particular, failures to accurately predict mutation probabilities for specific genes in specific samples can skew the expected mutation distribution, thus confounding the results. Examples including AID-related somatic hypermutation that can occur sporadically through a cohort[40,41] as well as false somatic mutation calls such as misclassified germline variants, which will lack correlation with the modelled selectively neutral distribution, thus leading to false positives. This again underscores the value of having access to independent metrics with different strengths and weaknesses.

Although we have established the usefulness of the principle, several refinements can be considered, including improvements to the mutational model. As indicated above, compared to frequency-based methods, accurate modelling of mutational activities in individual samples may be particularly important. In theory, WGS is therefore more suitable as it allows better modelling of sample-specific mutational signatures, although in practice we demonstrate good performance using WXS data. WGS data could also potentially allow local empirical estimation of mutation rate for each gene and sample, which may improve modelling, but this is currently not implemented. In practice, as power depends on the number of samples and mutations, the benefits of WGS may not always outweigh the downsides of WXS. Formal consideration of differences in cohort distribution between synonymous and non-synonymous mutations, expected to differ with respect to selective pressure in driver genes, or consideration of functional impact scores or positional effects in a similar manner, as well as consideration of indels, represent other avenues for improvement.

In conclusion, we describe a selection metric with orthogonal properties compared to current methods, including resilience to model inaccuracies that have confounded recent studies. We expect this to prove useful for finding new drivers and eliminating false positives. The principle has been implemented in the SEISMIC software (https://github.com/larsson-lab/SEISMIC).

## Methods

### SEISMIC implementation

The method, named SEISMIC (Selection Evidence Inferred from Sample-specific Modelling In Cancer), takes SNV mutation calls and regions to be analysed (e.g. coding sequences in genes or promoter regions) as input data.

Mutational models are created by calculating the frequency of each mutation type (i.e. the trinucleotide context and the variant nucleotide), normalised by the trinucleotide count in the region covered by the mutation dataset (whole genome for WGS data, and CDS regions for WXS data). For WGS data, the models are tumour-specific. For WXS data, the process is instead done on a cohort level due to the low number of mutations, and then normalised for each tumour by that tumour's total mutational burden, resulting in a model with a uniform trinucleotide signature, but with the amplitude varying between tumours. When using WXS data to analyse multiple cancer types together, such as in pan-cancer analyses, this cohort signature procedure is done per cancer type. This means that for WXS analyses of a single cancer type, only mutational burden affects the difference

between patients' mutational probabilities, whereas in analysis of multiple cancer types signatures of the cohorts have an impact on that difference.

If the test is performed on CDS regions using default settings, the mutations are annotated with the effects they cause, and synonymous mutations are disregarded as selection is not expected to act upon them (which effects to include/discard is configurable). For each tested region (e.g. a gene or promoter), the mutational model is applied to calculate the expected number of mutations (non-synonymous only in coding regions by default), and the probability in each tumour of observing at least one mutation (referred to below as the tumour being mutated, with respect to a specific gene or region). To remove recurrence effects from the test, this is then scaled so that the expected number of mutated tumours is equal to the actual observed number of mutated tumours for each gene, by scaling the expected number of mutations in each tumour uniformly. This step is required for the later likelihood determination of real and simulated cohorts, as the probability of observing one or more mutations in a tumour does not scale linearly with the expected number of mutations, necessitating model scaling. In essence, we are modelling inter-tumour differences in mutation rates, but discarding information about recurrence by scaling the expectations to match the observed recurrence.

For a given gene/region, the method then simulates a large number of cohorts (10,000 in this study) with the same number of mutated tumours as the actual cohort, by randomly distributing mutations among the tumours without replacement. This determines which tumours would be expected to contain mutations in the tested region in the absence of selection, using a mutation rate that is the same as in the observed data. As a result of this scaling, common mutation rate covariates such as replication timing or chromatin accessibility, which may differ from gene to gene (or region to region) but are typically not applied on a per-sample basis, are rendered irrelevant. Furthermore, using this procedure, excess mutation burden alone in the observed data will not be enough to produce a deviation from the modelled distribution, which is by design to introduce resilience to incorrect mutation rate estimates and to increase independence compared to frequency-based methods.

For each simulation, the likelihood of a specific combination of mutated tumours arising from the mutational model is calculated based on the modelled tumour-specific mutation probabilities. A gamma distribution is then fit to the distribution of simulated likelihood values, allowing for a one-sided test of whether the likelihood of the observed combination of mutated tumours for a given gene/region is unexpected relative to those of the simulated outcomes, thus indicating positive selection assuming the mutational model is accurate. This is the reported $p$-value from the method. The false discovery rate ($q$-value) is also reported, determined by adjusting the $p$-value for all the genetic regions included in a SEISMIC test using the Benjamini-Hochberg method.

In addition to assessing significance for cohort mutational skew, the method can also optionally report frequency statistics, thus indicating whether the number of mutations is elevated relative to the trinucleotide-based background model. This feature is not used in this paper.

### Other driver detection tools

ActiveDriverWGS:[9] Version 1.1.1. was installed in R from CRAN. Promoter regions were analysed using the full WGS somatic SNV mutation dataset (both described above), with no additional parameters. dNdScv:[5] Version 0.0.1.0 was installed in R from CRAN, and run with UCEC WXS hg19 SNVs with no additional parameters. Gene definitions intrinsic to dNdScv were used. MutSigCV:[4] Version 1.3.5 was run on the GenePattern[42] server, with the hg19 genome build, the default gene covariates file, and the supplied territory file directing MutSigCV to assume full exome coverage. UCEC WXS SNVs, annotated with

mutation effects and the standard mutation categories, were analysed in MutSigCV's internal definition of genes. MutPanning:[6] Version 2 was run on the GenePattern[42] server, with default parameters and using UCEC WXS hg19 SNVs.

## TCGA WXS somatic mutations

Whole-exome sequencing somatic mutation calls from TCGA based on VarScan2[43] and Mutect2[44] were downloaded from the NCI Genomic Data Commons (GDC) portal[45], and intersected to produce a high confidence set of somatic mutations. Mutations were further filtered to exclude any non-SNV mutations. For compatibility with other driver tools, genomic coordinates were converted from hg38 to hg19 using LiftOver[46], though the original hg38 coordinates were used with our method. The COAD and READ cancer types were combined to form a colorectal cancer (CRC) dataset. For each cohort analysis, tumours that contained at least 10 % of mutations in the cohort were removed. In each cancer type, the analysis was restricted to genes having at least 3 mutations. Cancer types with fewer than 1000 genes fulfilling this criterion were excluded from analysis, to avoid issues with cancer gene enrichment analysis, leaving BLCA, BRCA, CESC, CRC, ESCA, GBM, HNSC, LIHC, LUAD, LUSC, OV, SKCM, STAD, and UCEC. These cancer cohorts were analysed with our method as described above. False discovery rates ($q$-values) for the tests were calculated based on all genes included for each individual cohort. In the pan-cancer analysis, no cancer type exclusions were made, thereby including tumours from ACC, BLCA, BRCA, CESC, CHOL, CRC, DLBC, ESCA, GBM, HNSC, KICH, KIRC, KIRP, LAML, LGG, LIHC, LUAD, LUSC, MESO, OV, PAAD, PCPG, PRAD, SARC, SKCM, STAD, TGCT, THCA, THYM, UCEC, UCS, and UVM. Driver support tiers for all mutations in genes marked as significant by SEISMIC were attained from version 96 of the Cancer Mutation Census[22], available at https://cancer.sanger.ac.uk.

## WGS somatic mutations

For analysis of non-coding regions in melanoma, we used somatic SNV calls from melanoma WGS data. These consisted of mutations from the Australian Melanoma Genome Project (AMGP)[17] downloaded from the International Cancer Genome Consortium's (ICGC) database[47] pooled with whole-genome calls from the TCGA melanoma cohort[18] called with SAMtools mpileup (options -q1 and -B) and VarScan (strand filter = 1) and post-filtered to retain only variants with variant allele frequency > 0.2 and having a minimum coverage of eight and six reads in the normal and tumour samples, respectively, as previously described[48]. Where multiple samples were available for a patient, only the sample with the highest median variant allele frequency was retained, resulting in a total of 221 unique tumours[49]. Population variants (dbSNP v138) as well as any non-SNV mutations were excluded. In CDS-based analysis in Fig. 2b, c, the TCGA subset of these tumours ($n$ = 38) were excluded in order to make the cohort independent of the TCGA WXS-based coding sequence analysis.

Publicly available Pan-cancer Analysis of Whole Genomes (PCAWG)[25] consensus SNVs were obtained from ICGC[47] and combined with the controlled access US portion of the cohort (downloaded via Bionimbus), for a total of 2658 tumours in the following cancer types: Biliary-AdenoCA, Bladder-TCC, Bone-Cart, Bone-Epith, Bone-Leiomyo, Bone-Osteosarc, Breast-AdenoCa, Breast-DCIS, Breast-LobularCa, Cervix-AdenoCA, Cervix-SCC, CNS-GBM, CNS-Medullo, CNS-Oligo, CNS-PiloAstro, ColoRect-AdenoCA, Eso-AdenoCa, Head-SCC, Kidney-ChRCC, Kidney-RCC, Liver-HCC, Lung-AdenoCA, Lung-SCC, Lymph-BNHL, Lymph-CLL, Lymph-NOS, Myeloid-AML, Myeloid-MDS, Myeloid-MPN, Ovary-AdenoCA, Panc-AdenoCA, Panc-Endocrine, Prost-AdenoCA, Skin-Melanoma, Stomach-AdenoCA, Thy-AdenoCA, and Uterus-AdenoCA.

## Genomic region definitions

RefSeq[50] gene definitions for the GRCh37.p13 and GRCh38.p13 genome assemblies were downloaded from the UCSC ftp server. To limit transcripts to one per gene, only curated RefSeq Select versions were kept. Genes not on chromosomes 1–22 were filtered out. A small number of genes whose CDS lengths were not evenly divisible by 3, and thereby not consisting only of full codons, were excluded. This resulted in a set of 18,293 genes for analysis in hg38, and 18,271 genes in hg19. Gene promoters were defined as the 500 bp immediately upstream of the annotated transcription start site. Version 96 of the Cancer Gene Census[22], available at https://cancer.sanger.ac.uk, was used to define canonical cancer genes.

## Hypermutated tumours

For the performance comparison of including hypermutated tumours (Supplementary Fig. 7), SEISMIC and dNdScv were run on the full TCGA UCEC cohort (529 tumours), as well as with hypermutated tumours removed. Tumours with at least 2000 mutations were classified as hypermutated, numbering 61 tumours. dNdScv was run with max_muts_per_gene_per_sample = Inf, max_coding_muts_per_sample = Inf to disable internal filtering of hypermutated tumours.

## Reporting summary

Further information on research design is available in the Nature Portfolio Reporting Summary linked to this article.

## Data availability

TCGA WXS mutation calls were downloaded from the NCI GDC portal (https://portal.gdc.cancer.gov/). WGS-based melanoma mutation calls from AMGP, as well as publicly available WGS pan-cancer mutation calls from PCAWG, were attained from the ICGC database (https://dcc.icgc.org/). The restricted TCGA portion of the PCAWG dataset was downloaded via Bionimbus (https://bionimbus-pdc.opensciencedatacloud.org/). TCGA WGS melanoma mutation calls were based on alignments downloaded from cgHub (cgHub is no longer available; TCGA WGS data is now accessible through GDC). Researchers need to apply for access to TCGA WGS data to the TCGA Data Access Committee (DAC) via dbGaP (https://dbgap.ncbi.nlm.nih.gov). The study makes use of Cancer Gene Census (v96) and Cancer Mutation Census (v96) data, downloaded from https://cancer.sanger.ac.uk, as well as RefSeq gene definitions (GRCh37.p13 and GRCh38.p13), downloaded from the UCSC ftp server (https://genome.ucsc.edu/). The MD-2/LPS interface protein structure (PDB ID 3FXI) was accessed at RCSB Protein Data Bank (https://www.rcsb.org/). Source data are provided with this paper.

## Code availability

The SEISMIC software is freely available for non-commercial use at https://github.com/larsson-lab/SEISMIC. The version of the software used in this paper is available at https://doi.org/10.5281/zenodo.7247905[51]

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

## Acknowledgements

The results published here are in whole or part based upon data generated by The Cancer Genome Atlas pilot project established by the NCI and NHGRI, as well as ICGC. Information about TCGA and the

investigators and institutions who constitute the TCGA research network can be found at http://cancergenome.nih.gov. We are most grateful to the patients, investigators, clinicians, technical personnel, and funding bodies who contributed to TCGA and ICGC, thereby making this study possible. E.L. is supported by grants from the Knut and Alice Wallenberg Foundation, the Swedish Medical Research Council and the Swedish Cancer Society.

## Author contributions

Study design: M.B. and E.L. Concept: E.L. Data analysis: M.B. Software design: M.B. Manuscript: M.B. and E.L.

## Funding

## Competing interests

The authors declare no competing interests.
