## [Peer Review File · Nature Communications]

REVIEWER COMMENTS

Reviewer #1 (Remarks to the Author): Expert in cancer genomics and melanomas

The paper by Bostrom and Larsson proposes a complementary method for the analysis of cancer genome data for driver gene discovery. It follows other approaches such as dNdScv and MutSig which were developed for the same purpose. The paper is very well written and clear and the code and Github information/page is well presented.

I have a number of specific comments/suggestions:

1. On the github page the authors indicate the method is pinned to hg19. This is somewhat surprising given that the current genome assembly is GRCh38. Can the authors clarify this and explain how their method will be future proofed?
2. For the sentence starting “This stood in contrast to the established passenger genes...” 4 genes are mentioned. For each of these genes there is some, albeit contradictory in some cases, evidence that they could be drivers and the “ground truth” that is cited is another study that also examined driver gene probabilities using the TCGA dataset. In the same way Bostrom and Larsson claim to identify a large number of new/novel genes. For UCEC for example a long list of candidate drivers is supplied – the functional data to examine the possible contribution of these genes (such as data from DepMap) is not considered. If these genes are “truly” drivers what experimental evidence that the authors can supply to support this?
3. Since the authors cite Ref17 – how well does their method compare? Further, the analysis in this paper should be considered: Bailey, M. H. et al. Comprehensive characterization of cancer driver genes and mutations. Cell 173, 371–385 (2018).
4. All of the genes in Figure 4 should be put in a supplementary table so that other investigators don't need to “screen scrape” them from the paper for future comparisons.

Reviewer #2 (Remarks to the Author): Expert in bioinformatics, driver discovery, and cancer genomics

Boström and Larsson describe a new mutation driver discovery tool, SEISMIC. SEISMIC identifies mutations under selection by calculating mutation probabilities taking into account TMB, DNA sequence, and trinucleotide context. This is done on a cohort level for exome data, but a mode for wgs is also available.

The method uses only nonsynonymous mutations to identify drivers and creates a background distribution from all the mutations. The manuscript is overall well written. The method description is very short and limits the ability to fully understand the impact of covariates on the model etc. Link to git repository is provided and I could make the example code work.

The approach is interesting, but also in many ways similar to the approach of dndscv, which corrects for a number of covariates and assesses differences in synonymous to non-synonymous mutations, without specific recurrence analysis. SEISMIC appears to represent a slightly different version of dndscv with some added advantages and disadvantages. SEISMIC does rely on nonsynonymous mutations to detect drivers, but not specifically dN/dS ratios, which could make it suitable in principle for non-coding regions.

In general, the manuscript has very limited scientific novelty, but the method could be a useful addition to the large plethora of recurrence-based driver discovery tools, however, it would need to be supplemented with several other such tools for specificity purposes, which the authors also acknowledge.

Specific comments.

- mutational processes primarily cause passenger mutations but will also cause driver mutations, and I'm worried that many true-positive (clustered) mutations are removed, due to this. How many presumably true drivers are identified by MutSigCV and/or dndscv but are missed by SEISMIC?

- what's the weight of different covariates for the discovery - e.g. GC bias, sequencing coverage etc for the sensitivity and specificity?

- A recent paper, published in Nat Comms, identified drivers associated with mutational signatures (PMID: 35013316). It would be interesting to compare the two methods since they should in principle be largely mutually exclusive.

- Fig 1 and 2 focus on melanoma, which is strongly influenced by UV-light signatures, and finding drivers that deviate from the CC>TT is therefore not surprising, but nevertheless a nice validation of the

approach. It is reported that the hotspot mutations GNAQ and KIT are missed by the other callers. It says in Supp Data 1 that 20 mutations were found for KIT - presumably drivers. In the dndscv table, it says four of these are synonymous mutations (and 16 are missense). How can they be drivers?

- Promoter mutations are specifically analyzed for melanoma, Fig 3, where the known TERT promoter mutation is present. I was surprised that a noncoding promoter analysis was not conducted on a larger (PanCancer) dataset, to discover putative novel non-coding mutations. The recent study by Dietlein et al 2022 would be a good comparison.

- Figure 4 suggests a fair sensitivity but lower specificity - based on the number of non-CGC genes nominated as driver mutations specifically by SEISMIC. Why is Bailey and Dietlein et al drivers not included in the "ROC-style" curve?

- How many consensus drivers (identified by the other tools) are missed by SEISMIC? Figure 4 should be supplemented with a list of known driver mutation sites and not just driver genes and further scrutiny of SEISMIC-private mutations should be performed for recurrence, variant effect prediction etc. It could be that any aberrant mutation is identified in driver genes, because it has an overall altered trinucleotide distribution of mutations, leading to false-positive calls.

Reviewer #3 (Remarks to the Author): Expert in cancer genomics and evolution, and driver discovery

In this clear and well-written manuscript, Boström and Larsson describe a novel method to identify signals of positive selection in cancer genomes. This new freely available algorithm, named SEISMIC, identifies genomic elements that deviate from the expected sample-specific mutational status across a cancer cohort. As the authors mention, and to the best of my knowledge, this is a novel signal that has not been previously exploited to identify genomic elements under selection in cancer and consequently is of great interest for the field.

I have some comments that could help to improve the current manuscript:

Major comments

1) Although the manuscript shows very nicely the utility of the new method to identify cancer drivers, the biological novelty of the findings is slightly limited. From my point of view, the method could be very useful for the analysis of cohorts of hypermutated tumors (e.g., MSI cancers), which are currently difficult to address by most state-of-the-art driver discovery methods. This or similar analysis could add biological novelty to the manuscript.

2) In the WXS melanoma analysis, I understand that the background model is built using the trinucleotide mutational probabilities (mutational profile) from the whole cohort. Considering that UV-exposed melanomas have a higher mutation burden and a different mutational profile to that of non-UV-exposed melanomas, I would expect the cohort-wise mutational profile to be dominated by UV-exposed melanomas. How is this impacting the identification of signals in the set of low mutation burden non-UV-exposed samples? In a more general sense, how does the heterogeneity of the cancer cohort affect the identification of drivers?

3) Some long genes are highlighted as new drivers in different cohorts. Generally, gene length is considered false positives in driver analysis, since long genes can accumulate more mutations than short genes just by chance. I find this is also applicable to the new method. If this were the case, I would suggest highlighting only those genes supported by additional evidence as in the case of KMT2B.

Minor comments

1) Although the authors provide enough information to understand the signal that their method identifies, in some parts of the text this signal is defined as a contrast to recurrence signals (e.g., “while largely ignoring recurrence and instead assessing [...]” in the last paragraph of the introduction). Since this can cause ambiguity, I believe it would be beneficial to define the new signal on its own.

2) I understand the methodology accounts for large-scale covariates because the modeling is carried out on a gene-wise basis. Is this correct? In case it is, it would be positive to include this information within the manuscript because it is an advantage compared to some driver discovery methods. Similarly, it is also worth highlighting that the method can be applied to coding and non-coding regions of the genome.

3) The code repository is well documented, however the expected output from the demo is missing.

Point-by-point response, NCOMMS-22-13174-T

Dear reviewers,

We would like to sincerely thank you for your useful comments and for taking the time to evaluate our manuscript. As detailed further below, we have made a number of major improvements to the SEISMIC tool and the manuscript, including added support for the hg38 genome assembly and inclusion of a pan-cancer screen of promoter mutations based on PCAWG data.

Some additional improvements were also made during the revision process:

- Simplified mutation file formats (easier proprietary format plus MAF support).
- All processing is now self-contained in SEISMIC.R
- Improved parallelisation, added progress bar and much lower RAM usage.
- Plotting support added, for select genes or all genes below a p -value cutoff. These plots (essentially the same as presented in the supplement) may help in further assessing the plausibility of significant genes.
- Bundled gene definitions changed to RefSeq Select, as opposed to our previous homebrewed solution. By default, we now limit analysis to genes in chr1-22, to avoid issues with special handling of sex chromosome mutations (gene definitions including chrX/Y are also provided).
- Fixed a minor bug impacting the background model in WXS runs with more than one cancer type.
- In **Fig. 4**, genes that were significant in multiple cancer types in Dietlein et al were previously always marked as only significant in another cancer type (e.g., *NRAS* in SKCM). This has been fixed.

These changes essentially do not alter the results and conclusions, but all figures and tables have nevertheless been regenerated.

Below, reviewer comments are shown in **red bold** while our replies are shown in black with manuscript excerpts shown in *italic*.

Reviewer #1 (Remarks to the Author): Expert in cancer genomics and melanomas

The paper by Bostrom and Larsson proposes a complementary method for the analysis of cancer genome data for driver gene discovery. It follows other approaches such as dNdScv and MutSig which were developed for the same purpose. The paper is very well written and clear and the code and Github information/page is well presented.

We are glad to see these positive comments, and want to thank the reviewer for providing us with useful constructive feedback below.

I have a number of specific comments/suggestions:

On the github page the authors indicate the method is pinned to hg19. This is somewhat surprising given that the current genome assembly is GRCh38. Can the authors clarify this and explain how their method will be future proofed?

Pinning to hg19 was a result of the initial dataset the method was tested on, but we fully agree this limitation needed to be rectified. We have added support for hg38, which will also be the default assembly in the template config file going forward. Furthermore, we have redone all of our tests that rely on TCGA WXS data to use hg38 coordinates, and thereby removed the necessity of using liftOver for the mutation data for SEISMIC in the paper. While this had only a minor impact on the output and while the main results and conclusions thus remain the same, we have updated all figures and tables to reflect these changes.

When a future assembly version comes along, we may either add that as a third option, or we may add a more general approach for other assemblies, depending on what seems appropriate.

For the sentence starting “This stood in contrast to the established passenger genes...” 4 genes are mentioned. For each of these genes there is some, albeit contradictory in some cases, evidence that they could be drivers and the “ground truth” that is cited is another study that also examined driver gene probabilities using the TCGA dataset. In the same way Bostrom and Larsson claim to identify a large number of new/novel genes. For UCEC for example a long list of candidate drivers is supplied – the functional data to examine the possible contribution of these genes (such as data from DepMap) is not considered. If these genes are “truly” drivers what experimental evidence that the authors can supply to support this?

Regarding putative driver events in these four genes, we agree in principle, although we also think it's reasonably safe to assume that the mutations observed in genes such as *TTN* are generally passenger events in most cancers. We have therefore modified this sentence to say “*likely passenger genes*”. Furthermore, we have now added a supplementary analysis (**Supplementary Fig. 2**) with 20 randomly selected non-CGC genes with large CDSs and introns in order to ensure sufficient mutations for visualisation in the heatmap. The genes consistently show mutation patterns comparable to *TTN*, *PCLO*, *DNAH7*, and *MUC17*, further supporting that the mutations in these genes are indeed generally passenger events. It should also be noted that the mutations in all of these genes closely follow the expected pattern under neutral selection as simulated by SEISMIC.

In contrast, the genes highlighted for UCEC in **Fig. 4a** show significant deviations from this expected pattern. At the same time, no model is perfect and, similar to other methods, some of our positive findings are likely to have arisen due to model imperfections rather than actual positive selection. We do believe that more cautious wording is warranted in relation to new candidate drivers (such as the ones highlighted in UCEC) to avoid giving the impression that we believe all of these genes are drivers, and have rephrased the text accordingly:

- When presenting our highlighted genes: “*While these should be treated with caution, some were plausible true positives.*”
- The paragraph about long genes has been significantly toned down, with similar mention of potential false positives: “*While caution is warranted as these two CGC genes have previously been suggested to represent false positives...*”

The previously highlighted large gene *KMT2B* was removed due to annotation issues (we found that it was absent from the analyses in several of the other studies). Instead we choose to highlight another gene from **Fig. 4a**, *TLR4*, in the Results text. *TLR4* has been proposed to contribute to gastric cancer by mediating an inflammatory response to bacterial LPS (Yuan et al, *Cell Death Dis.* 2013), and SEISMIC identified mutational skew in *TLR4* specifically in STAD. Additional support for a driver function is given by the fact that a subset of the mutations in *TLR4* cluster around the LPS recognition site in this protein (**Supplementary Fig. 8-9**, the latter being a partial view of the TLR4 3D structure with mutated amino acids at the LPS binding site indicated). This is in agreement with an earlier study, which shows many such mutations modulate the response to LPS in a reporter system (Elliott et al, *Plos Genetics* 2017).

We agree that DepMap is a useful resource, and for many of the highlighted genes there are indeed growth effects. This includes *ING1*, a known inhibitor of cell growth; the inhibition of which is associated with modest but consistent increase in proliferation in DepMap cells lines:

However, the relevance of many of the findings may in practice be difficult to confirm using cell line proliferation data such as what is available in DepMap, as more complex

interactions with the tumour environment may be involved (as in the case of e.g. *TLR4*), and as such we have not included DepMap results in the study. While some of our findings may plausibly form the basis of more detailed experimental follow-ups, we feel that this is beyond the scope of this computational study.

Since the authors cite Ref17 – how well does their method compare? Further, the analysis in this paper should be considered: Bailey, M. H. et al. Comprehensive characterization of cancer driver genes and mutations. Cell 173, 371–385 (2018).

We agree that MutPanning is an interesting tool. A comparison between genes identified by SEISMIC and MutPanning results from Dietlein et al. (Ref17) is included in **Fig. 4a**. We have now also added a direct performance comparison with MutPanning by applying this tool, as well as SEISMIC, to the same UCEC mutation dataset. This is shown in **Fig. 4b-c**. For this cancer type, we find that MutPanning and SEISMIC perform similarly both in terms of sensitivity and specificity. Furthermore, we have added columns in **Supplementary Data 1** to indicate which of the genes analysed by SEISMIC are found by MutPanning in the Dietlein et al study, as well as three other studies indicated in **Fig. 4a**.

We also compare our results against those from Bailey et al in **Fig. 4a**, as their results represent an excellent set of consensus cancer genes. A direct comparison with the mutation dataset used in our paper (as in **Fig. 4b-c** for MutSigCV, dNdScv and now MutPanning) is not possible however, as Bailey et al.'s results are based on results from some 20 tools, and software to repeat that analysis is not available. However, all SEISMIC-analysed genes, significant or not, have now been annotated to indicate whether they are reported by Bailey et al. in **Supplementary Data 1**, as mentioned above.

All of the genes in Figure 4 should be put in a supplementary table so that other investigators don't need to "screen scrape" them from the paper for future comparisons.

We agree this should be easily accessible. This information is now available in **Supplementary Data 1**, which contains a column indicating all genes shown in **Fig. 4**. We have also added columns to this table indicating overlaps with the other cancer driver catalogues to which we compare our results.

Reviewer #2 (Remarks to the Author): Expert in bioinformatics, driver discovery, and cancer genomics

Boström and Larsson describe a new mutation driver discovery tool, SEISMIC. SEISMIC identifies mutations under selection by calculating mutation probabilities taking into account TMB, DNA sequence, and trinucleotide context. This is done on a cohort level for exome data, but a mode for wgs is also available. The method uses only nonsynonymous mutations to identify drivers and creates a background distribution from all the mutations. The manuscript is overall well written. The method description is very short and limits the ability to fully understand the impact of covariates on the model etc. Link to git repository is provided and I could make the example code work.

We want to thank the reviewer for evaluating our manuscript and for pointing out the need for a more detailed methods description. We agree that some more in-depth description of the various steps were warranted, and have now expanded the text in several places, including the impact of covariates in Methods. As an example, a more in-depth description about simulations and covariates has been included:

“For a given gene/region, the method then simulates a large number of cohorts (10,000 in this study) with the same number of mutated tumours as the actual cohort, by randomly distributing mutations among the tumours without replacement. This determines which tumours would be expected to contain mutations in the tested region in the absence of selection, using a mutation rate that is the same as in the observed data. As a result of this scaling, common mutation rate covariates such as replication timing or chromatin accessibility, which may differ from gene to gene (or region to region) but are typically not applied on a per-sample basis, are rendered irrelevant. Furthermore, using this procedure, excess mutation burden alone in the observed data will not be enough to produce a deviation from the modelled distribution, which is by design to introduce resilience to incorrect mutation rate estimates and to increase independence compared to frequency-based methods.

For each simulation, the likelihood of a specific combination of mutated tumours arising from the mutational model is calculated based on the modelled tumour-specific mutation probabilities. A gamma distribution is then fit to the distribution of simulated likelihood values, allowing for a one-sided test of whether the likelihood of the observed combination of mutated tumours for a given gene/region is unexpected relative to those of the simulated outcomes, thus indicating positive selection assuming the mutational model is accurate. This is the reported p-value from the method.”

As changes have been made throughout Methods, we are not quoting everything here. Please also see the next point regarding the general explanation of the SEISMIC procedure.

The approach is interesting, but also in many ways similar to the approach of dndscv, which corrects for a number of covariates and assesses differences in synonymous to non-synonymous mutations, without specific recurrence analysis. SEISMIC appears to represent a slightly different version of dndscv with some added advantages and disadvantages. SEISMIC does rely on nonsynonymous mutations to detect drivers, but not specifically dN/dS ratios, which could make it suitable in principle for non-coding regions.

We agree that the SEISMIC principle can be suitable for non-coding alterations: for example, we show that the lack of reliance on burden/frequency can make it less sensitive to confounding local mutational phenomena. The comment about similarity with dNdScv made us realise that the general description of our method, e.g. in the introduction, may not have been clear enough, as we believe the two methods are indeed based on very different principles. It should be noted that dNdSCV, as well as functional impact-based tools, can be considered to be frequency-based: these assess whether there is an excess of impactful mutations (e.g. non-synonymous or high predicted impact) compared to what is randomly expected (as estimated e.g. from synonymous mutations counts). In contrast, SEISMIC does

not care about the frequency but only *where* in the cohort mutations occur (the background model is adjusted to always produce the same number of mutations as is observed).

The last paragraph of the introduction has been rephrased to better clarify this, also defining the term “cohort mutational skew” to be used as a contrast to other approaches. This paragraph now reads:

“Similar to many frequency-based approaches, we use a background model that considers trinucleotide-based signatures¹⁶. However, recurrence is ignored and instead we assess statistically whether the pattern of mutated cases across a cohort for a given gene deviates from the expectation under neutral selection (“cohort mutational skew”). While conceptually different, an analogy can be made to how other methods instead test whether mutations show an unlikely positional or functional impact score distribution^{3,7,8}. Our results show that this approach can be less sensitive to flaws in the background model and may therefore better handle as yet unidentified mutational heterogeneity effects. Being based on a different principle, the method provides a much-needed complement that may help strengthen or weaken the verdict of existing tools.”

Together with clarifications in the Methods description (see point 1 above), we hope that the principle for detecting selection signals that underlies SEISMIC, and how it differs compared to widely used methods such as dNdScv, is now clearer.

In general, the manuscript has very limited scientific novelty, but the method could be a useful addition to the large plethora of recurrence-based driver discovery tools, however, it would need to be supplemented with several other such tools for specificity purposes, which the authors also acknowledge.

While SEISMIC is not necessarily more sensitive or more specific, we believe it does differ considerably from the commonly used recurrence-based methods in terms of the underlying basic principle for detecting selection signals. We therefore agree that it constitutes a useful complement to the existing toolbox for improved gene prioritisation in cancer studies.

Specific comments.

Mutational processes primarily cause passenger mutations but will also cause driver mutations, and I’m worried that many true-positive (clustered) mutations are removed, due to this. How many presumably true drivers are identified by MutSigCV and/or dndscv but are missed by SEISMIC?

First, we need to clarify that our method does not explicitly care about whether mutations are clustered or not. Whether clustered (e.g. caused by a local mutational process) or spread out across a gene region, our tool will assess whether the *distribution of mutations across the patient cohort* (rather than the frequency) is “unexpected” i.e. indicative of selective pressure acting upon them.

As such, a driver mutation being caused by a local mutational process, e.g. local clustered C>T mutations in a dipyrimidine context in skin melanoma, are not precluded from being identified by SEISMIC. If the mutations are under positive selection, they should be

overrepresented among patients with lower mutation probability (i.e. lower UV mutation burden). As an example, *MAP2K1* is one of our skin melanoma results. Of the 20 mutations that occur in recurrently mutated positions (13 P124S, 4 P124L, 2 E203K, and 1 E203V), 19 are compatible with mutagenesis through UV-induced CPD formation (C>T in a dipyrimidine), and there is clear skew of these mutations toward low-burden patients (see the attached SEISMIC-generated plot). Similarly, clustered C>T mutations in the *TERT* promoter show strong deviation from the expected cohort distribution (**Fig. 3**), in agreement with its well-established role in cancer. This can be compared with the *RPL13A* promoter (**Fig. 3**), which is also dominated by clustered UV-induced mutations, but which do not show the same skew, presumably due to lack of positive selection (*RPL13A* is a house-keeping gene not listed in the cancer gene census).

There are sure to be many true drivers missed by SEISMIC, similar to other tools, but we feel the important part of the message is the untapped potential of assessing the distribution of mutations among patients to evaluate selection.

While it is hard to compare gene-by-gene with other tools, as one cannot say with certainty what hits reported e.g. by MutSig are true or false positives, the general enrichment of known cancer genes among top reported hits can give a good hint about the sensitivity for detecting true driver genes. As can be seen in **Fig. 4c**, in the case of UCEC, SEISMIC reports 44 genes of which 26 are known CGC cancer genes (i.e. a considerable enrichment over the expected ~5%). The results from other tools when applied to the same data are fairly comparable: 24/34 for dNdScv, 13/17 for MutSigCV and 22/41 for MutPanning (which was added in revision). Overall, we interpret this to mean that SEISMIC exhibits a sensitivity that is comparable to established tools.

What's the weight of different covariates for the discovery - e.g. GC bias, sequencing coverage etc for the sensitivity and specificity?

Here we need to clarify that GC bias or sequencing coverage is not taken into account into our model. While this could be a good idea in the case of a frequency-based tool to improve estimation of expected frequencies, we do not necessarily think it would be useful in our case, since we do not consider the degree of recurrence when assessing signals of selection.

A recent paper, published in Nat Comms, identified drivers associated with mutational signatures (PMID: 35013316). It would be interesting to compare the two methods since they should in principle be largely mutually exclusive.

We presume that mutational signatures here refer to localised mutational processes. The results in the mentioned article seem to primarily represent localized AID-mediated somatic hypermutation in lymphoma, as well as APOBEC hotspots that can arise at hairpin structures and are generally considered to be passengers (see e.g. Buisson et al. *Science* 2019).

As clarified above, mutations at such sites are not necessarily excluded by our tool. We have now added a pan-cancer analysis of promoter mutations based on PCAWG WGS data (**Supplementary Data 6**). When comparing our results to theirs, we do indeed note that 3/6 of our hits overlap, one being the known cancer driver *TERT* and the other two being AID hotspots in *BCL7A* and *IGLL5*. We think it is likely that these represent false positive signals not under selection, and both are dismissed as such in the original PCAWG non-coding drivers paper (Rheinbay et al, *Nature* 2020), which we note in Results. We also elaborate further upon these signals in the discussion.

Fig 1 and 2 focus on melanoma, which is strongly influenced by UV-light signatures, and finding drivers that deviate from the CC>TT is therefore not surprising, but nevertheless a nice validation of the approach. It is reported that the hotspot mutations GNAQ and KIT are missed by the other callers. It says in Supp Data 1 that 20 mutations were found for KIT - presumably drivers. In the dndscv table, it says four of these are synonymous mutations (and 16 are missense). How can they be drivers?

“20” in our *KIT* result refers to the number of tumours with nonsynonymous mutations in the cohort. The total number of nonsynonymous mutations was 21. Additionally, there were 5

synonymous mutations for a total of 26 mutations. dNdScv reports slightly different numbers, since this tool excludes tumours with high mutation burden (without this filter their counts agree with ours).

We do not think that all of these mutations are drivers (the SEISMIC result simply suggests that selection may be acting on some of them). However, a few of them that are possible drivers based on the Cancer Mutation Census are highlighted in **Fig. 2d**. Some of the other nonsynonymous mutations may also be drivers, but most likely not all. We do not believe that the synonymous mutations are drivers.

We have changed the column name “Tumours with mutations” in Supplementary Data 1 to “Tumours with nonsynonymous mutations”, as the previous phrasing was misleading.

Promoter mutations are specifically analyzed for melanoma, Fig 3, where the known TERT promoter mutation is present. I was surprised that a noncoding promoter analysis was not conducted on a larger (PanCancer) dataset, to discover putative novel non-coding mutations. The recent study by Dietlein et al 2022 would be a good comparison.

We agree that a larger non-coding promoter analysis is a good idea. As stated above, we have now added a comprehensive analysis of gene promoters based on PCAWG WGS mutation data, testing individual cancer types separately as well as a combined pan-cancer cohort of ~2,600 samples. In this analysis, *TERT* still stands head and shoulders above other results. Five additional significant promoters were found, three of which we believe are likely false positives due to AID-mediated hypermutation (see above). The *POLR3E* promoter, also identified by Rheinbay et al. (*Nature* 2020, PCAWG non-coding mutation study) using the same dataset, remains as an interesting putative non-coding driver, as well as *HNRNPR* in uterine cancer, although unidentified mutational model bias naturally cannot be excluded. Dietlein et al identifies a large number of hits related to likely transcription-coupled mutagenesis in tissue-specific genes, and neither *POLR3E* nor *HNRNPR* are highlighted in their paper.

Figure 4 suggests a fair sensitivity but lower specificity - based on the number of non-CGC genes nominated as driver mutations specifically by SEISMIC. Why is Bailey and Dietlein et al drivers not included in the “ROC-style” curve?

Fig. 4b-c (which includes the ROC-style curve) shows a comparison of different tools, including SEISMIC, applied to the same exact dataset. We agree that Dietlein et al.’s tool (MutPanning) should have been included there from the start, and have amended the figure accordingly. Bailey et al. is a different matter as it doesn’t represent a specific tool (their results are from a combination ~20 tools), and as such cannot be applied to the same dataset.

How many consensus drivers (identified by the other tools) are missed by SEISMIC? Figure 4 should be supplemented with a list of known driver mutation sites and not just driver genes and further scrutiny of SEISMIC-private mutations should be performed for recurrence, variant effect prediction etc. It could be that any aberrant

mutation is identified in driver genes, because it has an overall altered trinucleotide distribution of mutations, leading to false-positive calls.

First we need to clarify that SEISMIC does not act at the level of individual mutations but rather assesses, for a given gene, whether the overall distribution of mutations across a cohort deviates from the expected pattern under neutral selection (as discussed above). Thus, specific driver mutation sites cannot be reported, as no distinction is formally made between driver and passenger events in a given gene. However, we acknowledge that more detailed information at the single mutation level can be useful when evaluating the results (previously only available in figures for select genes).

While adding figures for all identified genes would not be possible, we have now added detailed mutation data as a supplementary table (**Supplementary Data 7**) for all genes identified in the main pan-cancer screen. This table includes information about Cancer Mutation Census driver tiers, SIFT and PolyPhen impact predictions for all individual mutations, as well as the burden rank of the patient with the mutation.

Furthermore, we have added functionality to SEISMIC itself to help users assess the individual mutated sites. Plots can now be automatically generated, showing at what positions and in which patients mutations occur, their effect on protein output (missense/nonsense/synonymous), as well as recurrently mutated positions.

Example plot for *AKT1* in UCEC:

Reviewer #3 (Remarks to the Author): Expert in cancer genomics and evolution, and driver discovery

In this clear and well-written manuscript, Boström and Larsson describe a novel method to identify signals of positive selection in cancer genomes. This new freely available algorithm, named SEISMIC, identifies genomic elements that deviate from the expected sample-specific mutational status across a cancer cohort. As the authors mention, and to the best of my knowledge, this is a novel signal that has not been previously exploited to identify genomic elements under selection in cancer and consequently is of great interest for the field.

We thank this reviewer for these appreciative comments as well as the constructive feedback given below.

I have some comments that could help to improve the current manuscript:

Major comments

1) Although the manuscript shows very nicely the utility of the new method to identify cancer drivers, the biological novelty of the findings is slightly limited. From my point of view, the method could be very useful for the analysis of cohorts of hypermutated tumors (e.g., MSI cancers), which are currently difficult to address by most state-of-the-art driver discovery methods. This or similar analysis could add biological novelty to the manuscript.

This is a very interesting idea. We have tried analysing exclusively MSI cohorts, as well as only hypermutated tumours in a cohort, but did not obtain impressive results. However, since SEISMIC works by exploiting the difference in mutation probabilities between patients, hypermutated tumours should in theory be helpful as a contrast in cohorts that also contain tumours with lower mutation burdens. In other tools, hypermutated tumours may have an adverse effect on driver detection, and are therefore often filtered out.

We therefore tested running both SEISMIC and dNdScv on the TCGA UCEC cohort, both with and without hypermutated tumours, while disabling dNdScv's hypermutation filters (dNdScv was chosen because of the available parameters to control these filters). dNdScv's driver detection capabilities declined when including hypermutated tumours, while SEISMIC's improved. This is of course not how dNdScv is meant to be used, but it does support that SEISMIC can utilise hypermutated tumours to improve cancer gene detection. We have added this result to the supplement as **Supplementary Fig. 7** and have added a brief mention about this to Results:

*"Notably, inclusion of many hypermutated tumours, which may otherwise present a challenge for driver detection tools³², contributed positively to SEISMIC's ability to discriminate drivers from passengers in this cohort (**Supplementary Fig. 7, Supplementary Data 1, 8-10**)."*

2) In the WXS melanoma analysis, I understand that the background model is built using the trinucleotide mutational probabilities (mutational profile) from the whole cohort. Considering that UV-exposed melanomas have a higher mutation burden and a different mutational profile to that of non-UV-exposed melanomas, I would expect the cohort-wise mutational profile to be dominated by UV-exposed melanomas. How is this impacting the identification of signals in the set of low mutation burden non-UV-exposed samples? In a more general sense, how does the heterogeneity of the cancer cohort affect the identification of drivers?

It is true that potentially useful information is being discarded when not using patient-specific signatures, as we touched upon in the Discussion section. To investigate how big of an impact this has on the outcome, we have added an analysis where we ran SEISMIC on CDS regions using PCAWG WGS data, using both patient-specific mutational models and the cohort-based approach normally used for WXS data. We have included these results as **Supplementary Data 14-15**. The correlation between the results, as assessed by comparing *p*-values for individual genes, was generally very high (**Supplementary Fig. 12**)

with no obvious change in the prioritisation of cancer genes (**Supplementary Fig. 13**). The largest difference was seen in CRC ($r = 0.93$ based on log-transformed p -values). In melanoma, results were particularly consistent in-between the two models ($r=1.00$). Most likely, the impact of inter-patient signature differences is in many cases being overshadowed by the larger difference in mutation burden.

A new paragraph had been included at the end of Results, where this analysis is presented:

*“With WXS data, SEISMIC uses a simplified mutation model where trinucleotide signatures are determined per cancer type rather than per tumour. In analyses of single cancers, this essentially makes the mutation probability for a given gene and tumour proportional to the tumour’s mutation burden. Exome-based analysis of PCAWG WGS mutation calls allowed us to compare the two models using the same data. While significant hits expectedly were reduced compared to the larger WXS-based analysis, we found that results based on the simplified (WXS) model correlated strongly with those from the patient-specific (WGS) model in most cancer types (**Supplementary Fig. 12, Supplementary Data 14-15**), with little difference in cancer gene enrichment among top-scoring genes (**Supplementary Fig. 13**). In analyses of coding regions, the increased statistical strength provided by currently available WXS cohorts may thus often outweigh the benefits of more granular mutational modelling in WGS data.”*

3) Some long genes are highlighted as new drivers in different cohorts. Generally, gene length is considered false positives in driver analysis, since long genes can accumulate more mutations than short genes just by chance. I find this is also applicable to the new method. If this were the case, I would suggest highlighting only those genes supported by additional evidence as in the case of KMT2B.

In principle, SEISMIC should be less sensitive to the confounding effects of long genes, as the algorithm does not consider the absolute mutation burden (only the distribution of mutations across the cohort). However, caution may still be warranted: when the number of mutations is large, statistical power increases, which in turn can amplify imperfections in the underlying model (i.e. even a small deviation from the modelled distribution can become statistically significant). Even though some of the long genes we uncover appear to exhibit relatively convincing deviations from the expected distribution, we have decided to err on the side of caution and tone down our claims, specifically removing mention of long genes from the abstract, and moderating the text in the relevant section in the results.

However, we do feel that there is some merit to mentioning them, due to SEISMIC’s different approach compared to frequency-based tools. Should a long gene accumulate more mutations in a way that is confounding to frequency-based tools, those mutations would simultaneously have to skew toward low-probability patients to have the same effect on SEISMIC. While this could of course happen, we feel that the different approach offered by SEISMIC may be useful as a complement when evaluating these difficult genes. The (now moderated) paragraph about long genes now reads:

“SEISMIC also reported several large cancer genes not detected in the other studies, including CSMD3 (3,707 amino acids) in UCEC and STAD, and LRP1B (4,599 amino acids) in STAD and colorectal cancer (CRC; Fig. 4a). A low driver-to-passenger mutation ratio in

large genes could make selection signals challenging to detect using traditional methods, while SEISMIC would be more sensitive to a relatively small number of driver events if these deviate markedly from the modelled cohort distribution. While caution is warranted as these two CGC genes have previously been suggested to represent false positives⁴, our results may justify further investigation of putative driver roles.”

Regarding *KMT2B*, we have decided to exclude it from the gene highlights after noticing that the gene was in fact not represented in the gene annotations used in most of the other studies, likely because of gene symbol aliases (there is confusion regarding *MLL2* and *MLL4*, which are both aliases to *KMT2B* in NCBI GENE, in addition to several other symbols; all other genes highlighted in the text in our study were represented in the other studies).

Minor comments

1) Although the authors provide enough information to understand the signal that their method identifies, in some parts of the text this signal is defined as a contrast to recurrence signals (e.g, “while largely ignoring recurrence and instead assessing [...]” in the last paragraph of the introduction). Since this can cause ambiguity, I believe it would be beneficial to define the new signal on its own.

We thank the reviewer for this important comment. In response to this, we have rephrased the last paragraph of the introduction, to further clarify the underlying principle and how it differs from existing methods. We are also introducing the term “cohort mutational skew”, which is then used again in the following heading and throughout the results text. The paragraph now reads:

“Similar to many frequency-based approaches, we use a background model that considers trinucleotide-based signatures¹⁶. However, recurrence is ignored and instead we assess statistically whether the distribution of mutated cases across a cohort for a given gene deviates from the expectation under neutral selection (“cohort mutational skew”). While conceptually different, an analogy can be made to how other methods instead test whether mutations show an unlikely positional or functional impact score distribution^{3,7,8}. Our results show that this approach, which is applicable to both coding and non-coding regions, can be less sensitive to flaws in the background model and may therefore better handle as yet unidentified mutational heterogeneity effects. Being based on a different principle, the method provides a much-needed complement that may help strengthen or weaken the verdict of existing tools.”

2) I understand the methodology accounts for large-scale covariates because the modeling is carried out on a gene-wise basis. Is this correct? In case it is, it would be positive to include this information within the manuscript because it is an advantage compared to some driver discovery methods. Similarly, it is also worth highlighting that the method can be applied to coding and non-coding regions of the genome.

This is correct. The method considers each gene in isolation, and for a given gene, the background model is calibrated to produce the same number of mutations as is observed. The statistical evaluation is based solely on how the observed mutations are *distributed* across the cohort. The absolute mutation count/burden for each gene is therefore rendered

irrelevant, which essentially makes gene-to-gene large-scale covariates irrelevant (under the assumption that such covariates stay constant between patients).

We have clarified this in Methods. Excerpt:

“As a result of this scaling, common mutation rate covariates such as replication timing or chromatin accessibility, which may differ from gene to gene (or region to region) but are typically not applied on a per-sample basis, are rendered irrelevant. Furthermore, using this procedure, excess mutation burden alone in the observed data will not be enough to produce a deviation from the modelled distribution, which is by design to introduce resilience to incorrect mutation rate estimates and to increase independence compared to frequency-based methods.”

A clarification was also done in the first paragraph of Results:

*“Likelihood scores are used to statistically assess whether an observed outcome (cohort mutation pattern) deviates relative to a large number of simulated outcomes, which are drawn from the estimated probabilities while calibrating the recurrence to be the same as the observed data (**Fig. 1**). Information normally lost when mutations are summarized into counts is thus taken advantage of while mutation frequency is disregarded.”*

We have also highlighted in the Introduction the the method is equally applicable to coding and non-coding regions, as we agree this is is worth clarifying:

“Our results show that this approach, which is applicable to both coding and non-coding regions, can be less sensitive to flaws in the background model and may therefore better handle as yet unidentified mutational heterogeneity effects.”

3) The code repository is well documented, however the expected output from the demo is missing.

We agree that this should be available in GitHub, and have added the expected output of the demo to the repository.

REVIEWERS' COMMENTS

Reviewer #1 (Remarks to the Author):

The authors have thoughtfully revised the manuscript and answered my major technical questions/concerns. The paper reads very well and the work will be useful to the community.

Reviewer #2 (Remarks to the Author):

The authors have improved the manuscript and description of the method, and I have no major concerns remaining.

Reviewer #3 (Remarks to the Author):

I thank the authors for successfully addressing all my comments. I only have a minor comment: I believe the supplementary table containing SEISMIC results for CDS regions in TCGA UCEC including hypermutated tumours is missing. From my perspective, the changes in SEISMIC method (e.g., hg38 support) and the new analyses (e.g., promoter driver search and WXS vs WGS comparison) have improved the quality and the scope of the manuscript. I therefore recommend the manuscript for publication.

Point-by-point response

Once again we wish to thank all reviewers for their efforts and for giving us valuable feedback. Below we respond to the one remaining issue (reviewer 3).

Reviewer #1 (Remarks to the Author):

The authors have thoughtfully revised the manuscript and answered my major technical questions/concerns. The paper reads very well and the work will be useful to the community.

Reviewer #2 (Remarks to the Author):

The authors have improved the manuscript and description of the method, and I have no major concerns remaining.

Reviewer #3 (Remarks to the Author):

I thank the authors for successfully addressing all my comments. I only have a minor comment: I believe the supplementary table containing SEISMIC results for CDS regions in TCGA UCEC including hypermutated tumours is missing. From my perspective, the changes in SEISMIC method (e.g., hg38 support) and the new analyses (e.g., promoter driver search and WXS vs WGS comparison) have improved the quality and the scope of the manuscript. I therefore recommend the manuscript for publication.

The requested table is already part of Supplementary Data 1, as referenced in the manuscript, since the UCEC cohort in this analysis did not exclude any hypermutated tumours. However, we think this could be clearer, and have edited the heading in Supplementary Data 8 (the corresponding SEISMIC results excluding hypermutated tumours) to add the following text: “(results including hypermutated tumours are found in Supplementary Data 1)”